# Structural assessment of HLA-A2-restricted SARS-CoV-2 spike epitopes recognized by public and private T-cell receptors

Daichao Wu [1,2,3], Alexander Kolesnikov[1,3], Rui Yin [1,3], Johnathan D. Guest [1,3], Ragul Gowthaman [1,3], Anton Shmelev [4], Yana Serdyuk [4], Dmitry V. Dianov[4], Grigory A. Efimov [4], Brian G. Pierce [1,3 ✉] & Roy A. Mariuzza [1,3 ✉]

T cells play a vital role in combatting SARS-CoV-2 and forming long-term memory responses. Whereas extensive structural information is available on neutralizing antibodies against SARS-CoV-2, such information on SARS-CoV-2-specific T-cell receptors (TCRs) bound to their peptide–MHC targets is lacking. Here we determine the structures of a public and a private TCR from COVID-19 convalescent patients in complex with HLA-A2 and two SARS-CoV-2 spike protein epitopes (YLQ and RLQ). The structures reveal the basis for selection of particular TRAV and TRBV germline genes by the public but not the private TCR, and for the ability of the TCRs to recognize natural variants of RLQ but not YLQ. Neither TCR recognizes homologous epitopes from human seasonal coronaviruses. By elucidating the mechanism for TCR recognition of an immunodominant yet variable epitope (YLQ) and a conserved but less commonly targeted epitope (RLQ), this study can inform prospective efforts to design vaccines to elicit pan-coronavirus immunity.

[1] W.M. Keck Laboratory for Structural Biology, University of Maryland Institute for Bioscience and Biotechnology Research, Rockville, MD 20850, USA.
[2] Department of Histology and Embryology, Hengyang Medical School, University of South China, Hengyang, Hunan 421001, China. [3] Department of Cell Biology and Molecular Genetics, University of Maryland, College Park, MD 20742, USA. [4] National Research Center for Hematology, Moscow, Russia.
✉email: pierce@umd.edu; rmariuzz@umd.edu

Severe acute respiratory syndrome coronavirus 2 (SARS-CoV-2) is the virus responsible for the global coronavirus disease 2019 (COVID-19) pandemic[1–3]. Elucidating the mechanisms underlying the adaptive immune response to SARS-CoV-2 is crucial for predicting vaccine efficacy and assessing the risk of reinfection[4]. Neutralizing antibodies against SARS-CoV-2 have been studied extensively and are clearly protective, but may be short-lived and are not elicited in all infected individuals[5]. Emerging evidence indicates that T cells play a vital role in the clearance of SARS-CoV-2 and in formation of long-term memory responses to this virus[6–17].

The finding that SARS-CoV-2-specific T cell responses can be detected in the absence of seroconversion[6], along with the observation that agammaglobulinemia patients lacking B cells can recover from COVID-19[7], suggest that T cells may be able to mount an effective response against SARS-CoV-2 when antibody responses are inadequate or absent. Consistent with an important role of T cells in recovery from SARS-CoV-2 are findings that levels of activated T cells increase at the time of viral clearance[8] and that T cell lymphopenia is predictive of disease severity[9]. Another study showed that most COVID-19 convalescent patients (CPs) exhibit broad and robust SARS-CoV-2-specific T cell responses[10]. Additionally, those who manifest mild symptoms displayed a greater proportion of polyfunctional CD8$^+$ T cell responses compared with severely diseased cases, suggesting a role of CD8$^+$ T cells in reducing disease severity.

A beneficial role for T cells in combatting SARS-CoV-2 is in agreement with studies showing that both CD4$^+$ and CD8$^+$ cells are protective against the closely related SARS-CoV betacoronavirus (~80% sequence identity to SARS-CoV-2) that caused an atypical pneumonia outbreak in 2003[14–16]. Adoptive transfer of SARS-CoV-specific CD4$^+$ or CD8$^+$ T cells enhanced survival of infected mice, demonstrating that T cells are sufficient for viral clearance even in the absence of antibodies or activation of innate immunity[16]. Conversely, deep sequencing of >700 SARS-CoV-2 isolates revealed non-synonymous mutations in 27 MHC class I-restricted SARS-CoV-2 epitopes that may enable the virus to escape killing by cytotoxic CD8$^+$ T cells[17].

Compared to the relatively short-lived antibody response to SARS-CoV-2 and other coronaviruses[5,18], T cells may persist for longer periods of time. Memory T cells specific for SARS-CoV epitopes have been detected up to 11 years following infection[19,20]. In one study, SARS-CoV-2 memory CD8$^+$ T cells declined with a half-life of 3–5 months[21], that is similar to the half-life of memory CD8$^+$ T cells after yellow fever immunization[22]. In another study, SARS-CoV-2-specific T cell immunity was stable for 6 months[23]. Robust epitope-specific CD8$^+$ T cell responses have been detected in individuals immunized with the BNT162b2 mRNA vaccine, with magnitudes comparable to memory responses against CMV, EBV, and influenza virus[24].

Based on these and related findings, intensive efforts are underway to identify SARS-CoV-2 epitopes that elicit protective T cell responses against this virus and to delineate TCR repertoires specific for these epitopes[6,10,13,17,24–33]. T cell responses to ORFs encoding both structural (S, M, and N) and nonstructural (nsp3, 4, 6, 7, 12, and 13) proteins have been detected, with the S (spike) and N (nucleocapsid) proteins inducing the most robust CD8$^+$ T cell responses in most studies.

Four human coronaviruses are known to cause seasonal common cold respiratory infections: OC43, HKU1, NL63, and 229E. These viruses share partial sequence homology (~35%) with SARS-CoV-2. T cell responses to SARS-CoV-2 have been detected in 20–50% of pre-pandemic individuals, suggesting cross-reactive T cell recognition between common cold coronaviruses and SARS-CoV-2 that could potentially underlie some of the extensive heterogeneity observed in COVID-19 disease[27,30,34–36].

A wealth of structural information is now available on neutralizing antibodies from COVID-19 CPs bound to the SARS-CoV-2 spike trimer or receptor-binding domain (RBD) (>200 Protein Data Bank depositions), resulting in a highly detailed picture of the B cell response to this virus[37–40]. By contrast, no structural information is available for TCRs specific for SARS-CoV-2 (or any other coronavirus) bound to their peptide–MHC (pMHC) targets, despite the crucial role of T cells in orchestrating the antiviral response. Here we report crystal structures of one public and one private TCR (YLQ7 and RLQ3, respectively) from COVID-19 CPs in complex with HLA-A*02:01 and two S protein epitopes, corresponding to residues 269–277 (YLQPRTFLL; designated YLQ) and 1000–1008 (RLQSLQTYV; designated RLQ), that were found to elicit almost universal CD8$^+$ T cell responses in HLA-A2*02:01$^+$ CPs but not in healthy donors[26]. Public TCRs are observed in multiple unrelated individuals, whereas private TCRs are distinct between individuals. The YLQ epitope has been identified as immunodominant in multiple independent studies[33], including one involving the BNT162b2 mRNA vaccine[24]. For its part, the RLQ epitope, unlike YLQ, is conserved across human and zoonotic sarbecoviruses and is therefore a potential candidate for inclusion in a pan-sarbecovirus vaccine.

## Results

**Interaction of SARS-CoV-2-specific TCRs with spike epitopes and epitope variants**. Sequences of α and β chains for YLQ- and RLQ-specific TCRs were obtained from a previous study[26]. TCR α and β chains were paired based on their relative frequency and/or co-occurrence in samples obtained from the same patients. Sequences of TCRs selected for this study are provided in Supplementary Table 1. The epitope specificity of five of these TCRs was confirmed by staining with pMHC tetramers (Supplementary Fig. 1). RLQ3 utilizes gene segments TRAV16 and TRAJ39 for the α chain, and TRBV11-2 and TRBJ2-3 for the β chain, whereas YLQ7 utilizes TRAV12-2 and TRAJ30 for the α chain, and TRBV7-9 and TRBJ2-7 for the β chain. Of note, the α and β chain sequences of YLQ7 are identical to those reported for another YLQ-specific TCR that was identified independently using single-cell sequencing[32].

We used surface plasmon resonance (SPR) to measure the affinity of TCRs RLQ3 and YLQ7 for HLA-A2 loaded with RLQ or YLQ peptide (Fig. 1a, f). Recombinant TCR and pMHC proteins were expressed by in vitro folding from *E. coli* inclusion bodies. Biotinylated RLQ–HLA-A2 or YLQ–HLA-A2 was directionally coupled to a biosensor surface and increasing concentrations of TCR were flowed sequentially over the immobilized pMHC ligand. RLQ3 and YLQ7 bound RLQ–HLA-A2 and YLQ–HLA-A2 with dissociation constants ($K_D$s) of 32.9 μM and 1.8 μM, respectively (Fig. 1a, f). Importantly, these affinities are characteristic of TCRs with high functional avidity for microbial antigens, whose $K_D$s typically range between 1 μM and 50 μM[41]. In addition to RLQ3, we examined three other HLA-A*0201-restricted, RLQ-specific TCRs from COVID-19 CPs: RLQ5, RLQ7, and RLQ8 (Supplementary Table 1). These TCRs use completely different α/β chain pairs from RLQ3 (TRAV16/TRBV11-2), and from each other: RLQ5 (TRAV12-2/TRVB6-5), RLQ7 (TRAV38-2DV8/TRVB12-3), and RLQ8 (TRDV1/TRBV20-1). They bound RLQ–HLA-A2 with $K_D$s of 3.4 μM (RLQ5), 49.0 μM (RLQ7), and 9.7 μM (RLQ8) (Supplementary Fig. 2).

To test the functional properties of these TCRs, the Jurkat reporter cell line J76 TPR[42] was transduced with lentiviral vectors encoding the TCRs alongside CD8 co-receptor. Transduced cells were stimulated with the K562 cell line transgenic for

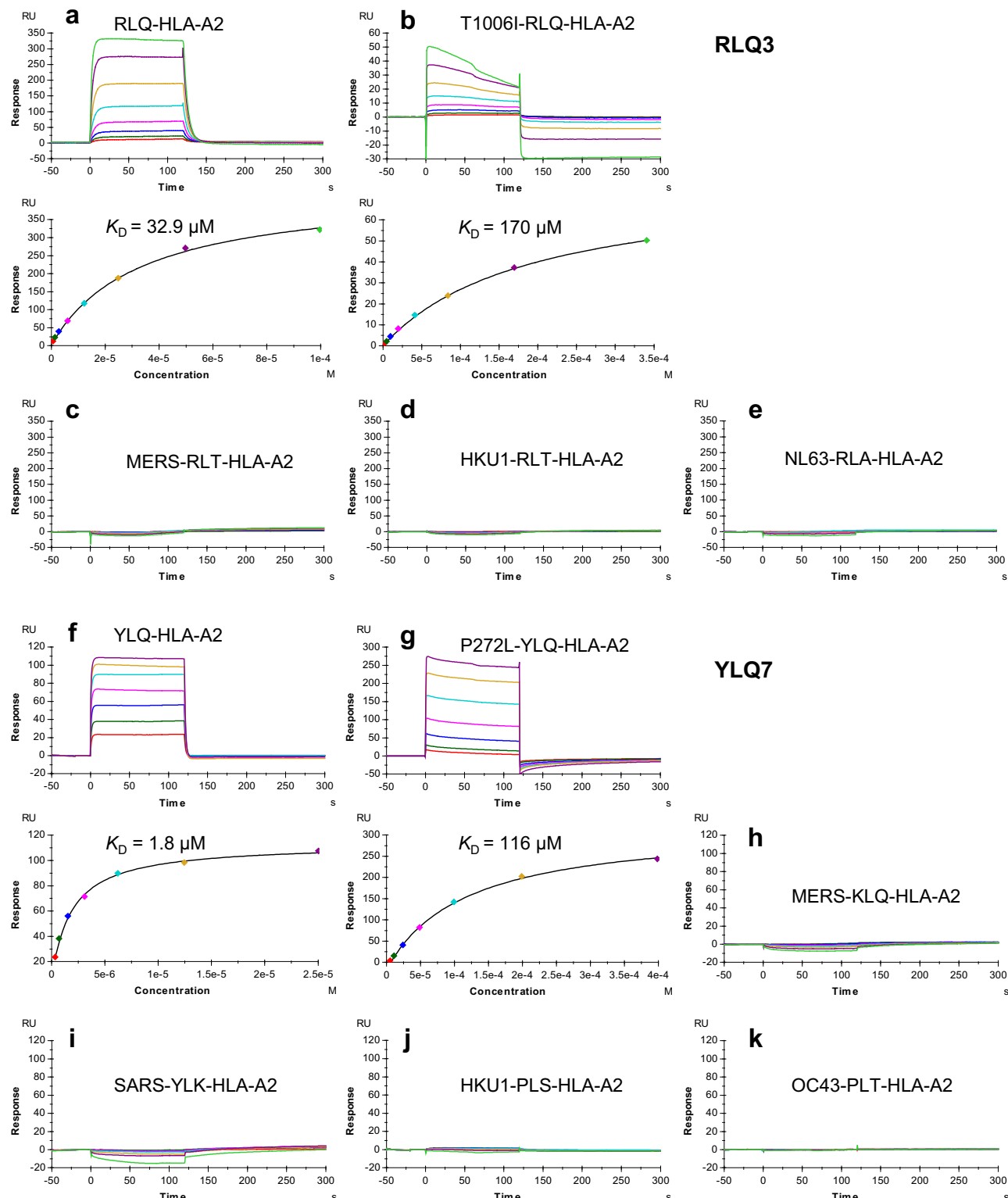

HLA-A*02:01 loaded with different RLQ or YLQ peptide concentrations. Activation was measured by expression of eGFP controlled by the NFAT promoter. In agreement with SPR results, all TCRs recognized the cognate epitopes, with functional avidities (EC$_{50}$) ranging from 0.07 μM for RLQ8 to 3.3 μM for RLQ3 (Fig. 2). In addition, we demonstrated that the public YLQ7 receptor with low micromolar affinity was only modestly dependent on the CD8 interaction (Supplementary Fig. 3), as shown previously for high affinity virus-specific TCRs[43,44].

We then tested SARS-CoV-2-specific TCRs RLQ3 and YLQ7 for cross-reactivity with other human coronaviruses (SARS-CoV, MERS, HKU1, OC43, and NL63) using peptides homologous to the RLQ and YLQ epitopes of SARS-CoV-2. These 9-mer peptides differ from RLQ at 4 or 5 positions and from YLQ at between 2 and 5 positions (Supplementary Table 2). SARS-CoV-2 and SARS-CoV share an identical RLQ epitope. We detected no interaction of RLQ3 and YLQ7 with any of these homologous peptides by SPR, even after injecting high concentrations of TCR

**Fig. 1 SPR analysis of SARS-CoV-2-specific TCRs binding to spike epitopes and epitope variants. a** (upper) TCR RLQ3 at concentrations of 0.78, 1.56, 3.12, 6.25, 12.5, 25.0, 50.0, and 100 μM was injected over immobilized RLQ–HLA-A2 (1200 RU). (lower) Fitting curve for equilibrium binding that resulted in a $K_D$ of 32.9 μM. **b** (upper) TCR RLQ3 at concentrations of 3.12, 6.25, 12.5, 25.0, 50.0, 100, 200, and 400 μM was injected over immobilized T1006I–HLA-A2 (1200 RU). (lower) Fitting curve for equilibrium binding that resulted in a $K_D$ of 170 μM. **c–e** TCR RLQ3 at concentrations of 0.39, 0.78, 1.56, 3.12, 6.25, 12.5, 25.0, 50.0, and 100 μM was injected over immobilized MERS-RLT–HLA-A2 (1600 RU), HKU1-RLT–HLA-A2 (700 RU), NL63-RLA–HLA-A2 (700 RU). **f** (upper) TCR YLQ7 at concentrations of 0.39, 0.78, 1.56, 3.12, 6.25, 12.5, and 25.0 μM was injected over immobilized YLQ–HLA-A2 (300 RU). (lower) Fitting curve for equilibrium binding that resulted in a $K_D$ of 1.8 μM. **g** (upper) TCR YLQ7 at concentrations of 6.25, 12.5, 25.0, 50.0, 100, 200, and 400 μM was injected over immobilized P272L–HLA-A2 (2000 RU). (lower) Fitting curve for equilibrium binding that resulted in a $K_D$ of 116 μM. **h–k** TCR pYLQ7 at concentrations of 0.39, 0.78, 1.56, 3.12, 6.25, 12.5, 25, and 50 μM was injected over immobilized MERS-KLQ–HLA-A2, SARS-YLK–HLA-A2, HKU1-PLS–HLA-A2, and OC43-PLT–HLA-A2, respectively (500 RU).

over the immobilized pMHC ligand (Fig. 1c–e, h–k). In a similar fashion, none of the epitopes derived from the seasonal coronaviruses, SARS-CoV, or MERS were able to stimulate transgenic TCR lines expressing RLQ3, YLQ7, RLQ5, RLQ7, or RLQ8, even at the highest peptide concentrations (Fig. 2). Thus, all TCRs examined are highly specific for SARS-CoV-2 and are unlikely to contribute to protection against these other coronaviruses, with the exception of SARS-CoV in the case of TCRs targeting the RLQ epitope, which is identical in SARS-CoV and SARS-CoV-2.

We also tested the ability of TCRs RLQ3 and YLQ7 to recognize two natural variants of the RLQ and YLQ epitopes found in the GISAID database (https://www.gisaid.org)[45]. The RLQ variant (designated T1006I) contains a threonine-to-isoleucine mutation at position 1006 (RLQSLQ**I**YV), while the YLQ variant (designated P272L) contains a proline-to-leucine mutation at position 272 (YLQ**L**RTFLL). These represent the most common mutations within these epitopes among SARS-CoV-2 spike glycoprotein sequences in the GISAID database; the low frequencies of those substitutions (T1006I: 0.04%; P272L: 0.56%) indicate that the RLQ and YLQ epitopes are well-conserved in SARS-CoV-2 (Supplementary Table 3). RLQ3 bound T1006I–HLA-A2 with a $K_D$ of 170 μM, representing a 5.2-fold affinity reduction relative to wild-type (Fig. 1b). YLQ7 bound P272L–HLA-A2 with a $K_D$ of 116 μM (Fig. 1g), corresponding to a much larger 64-fold reduction in affinity compared to the wild-type epitope. Similar to RLQ3, other RLQ-specific TCRs recognized the T1006I variant with lower affinity than the wild-type epitope, despite usage of unrelated α/β chain combinations: 23.0 μM for RLQ5 (6.8-fold reduction), 62.8 μM for RLQ7 (1.3-fold reduction), and >50 μM for RLQ8 (>5-fold reduction) (Supplementary Fig. 2).

Functional measurements ($EC_{50}$) revealed that different TCRs have a range of responses to the T1006I substitution, ranging from RLQ5 with a 32-fold functional avidity reduction (0.14 μM to 4.5 μM) to RLQ7 with a 3-fold functional avidity increase (0.28 μM to 0.09 μM) (Fig. 2). Overall, this indicated that on average the T1006I substitution was tolerated, in agreement with SPR data. In sharp contrast, YLQ7 was 189-fold less sensitive to the P272L variant than to the wild-type epitope ($EC_{50}$ = 76 μM for P272L versus 0.4 μM for wild-type YLQ). This drastic reduction in functional avidity makes unlikely efficient P272L recognition by primary T cells with the YLQ7 receptor and is consistent with the 64-fold loss of affinity as measured by SPR (Fig. 1). It is also consistent with a possible role of the P272L variant in evasion of the T cell response to SARS-CoV-2[46].

**Structures of RLQ–HLA-A2 and YLQ–HLA-A2.** We determined the structures of the RLQ–HLA-A2 and YLQ–HLA-A2 complexes to 2.81 and 2.07 Å resolution, respectively (Supplementary Table 4) (Fig. 3). Clear and continuous electron density extending the entire length of both MHC-bound peptides allowed confident identification of all peptide atoms (Supplementary Fig. 4). Both RLQ–HLA-A2 and YLQ–HLA-A2 crystals contain

two complex molecules in the asymmetric unit. The conformation of the RLQ peptide in the two RLQ–HLA-A2 complexes is nearly identical, with a root-mean-square difference (r.m.s.d.) of 0.14 Å for α-carbon atoms and 0.68 Å for all atoms (Fig. 3a). By contrast, the YLQ peptide adopts somewhat different conformations in the two YLQ–HLA-A2 complexes, with an r.m.s.d. of 0.52 Å for α-carbon atoms and 1.51 Å for all atoms (Fig. 3b). The largest differences occur in the central portion of the bound peptide, at P5 Arg and P6 Thr, whose α-carbons shift by 1.2 and 1.3 Å, respectively, and whose side chains rotate ~60° and ~90°, respectively, about the Cα–Cβ axis.

The RLQ and YLQ peptides are bound in conventional orientation with the side chains of P2 Leu and P9 Val/Leu accommodated in pockets B and F, respectively, of the peptide-binding groove (Fig. 3). These residues are among the most common at primary anchor positions P2 (Leu > Thr > Met ~ Val > Ile) and P9 (Val > Ile > Thr > Ala > Cys > Leu) and confer high affinity for HLA-A*02:01[47], in agreement with the immunogenicity of RLQ and YLQ in COVID-19 CPs[26,33]. In the RLQ–HLA-A2 complex, the solvent-exposed side chains of P1 Arg, P4 Ser, P5 Leu, P6 Gln, and P8 Tyr project away from the peptide-binding groove and compose a moderately featured surface for potential interactions with TCR (Fig. 3a). The YLQ epitope is more featured and comprises P1 Tyr, P4 Pro, P5 Arg, P6 Thr, P7 Phe, and P8 Leu, with the central P5 Arg residue contributing the most solvent-accessible surface area (146 Å²) (Fig. 3b).

**Overview of the RLQ3–RLQ–HLA-A2 and YLQ7–YLQ–HLA-A2 complexes.** To understand how TCRs RLQ3 and YLQ7 recognize their cognate S protein epitopes and to explain the effect of sequence differences or mutations in these epitopes on recognition, we determined the structures of the RLQ3–RLQ–HLA-A2 and YLQ7–YLQ–HLA-A2 complexes at 2.30 and 2.39 Å resolution, respectively (Supplementary Table 4) (Fig. 4a, d). The interface between TCR and pMHC was in unambiguous electron density in both complex structures (Supplementary Fig. 5). Both RLQ3 and YLQ7 dock symmetrically over RLQ–HLA-A2 and YLQ–HLA-A2 in a canonical diagonal orientation, with crossing angles of TCR to pMHC[48] of 36° and 41°, respectively (Fig. 4b, e), and with incident angles (degree of tilt of TCR over MHC)[49] of 18° and 4°, respectively. As depicted by the footprints of RLQ3 and YLQ7 on pMHC (Fig. 4c, f), both TCRs establish contacts with the N-terminal half of the peptide mainly through the CDR1α and CDR3α loops, whereas the CDR3β loop mostly contacts the C-terminal half.

**Interaction of TCR RLQ3 with HLA-A2.** Of the total number of contacts (54) that private TCR RLQ3 makes with HLA-A2, excluding the RLQ peptide, CDR1α, CDR2α, and CDR3α contribute 7%, 33%, and 11%, respectively, compared with 2%, 11%, and 36% for CDR1β, CDR2β, and CDR3β, respectively (Tables 1, 2). Hence, TCR RLQ3 relies on the somatically-generated CDR3α

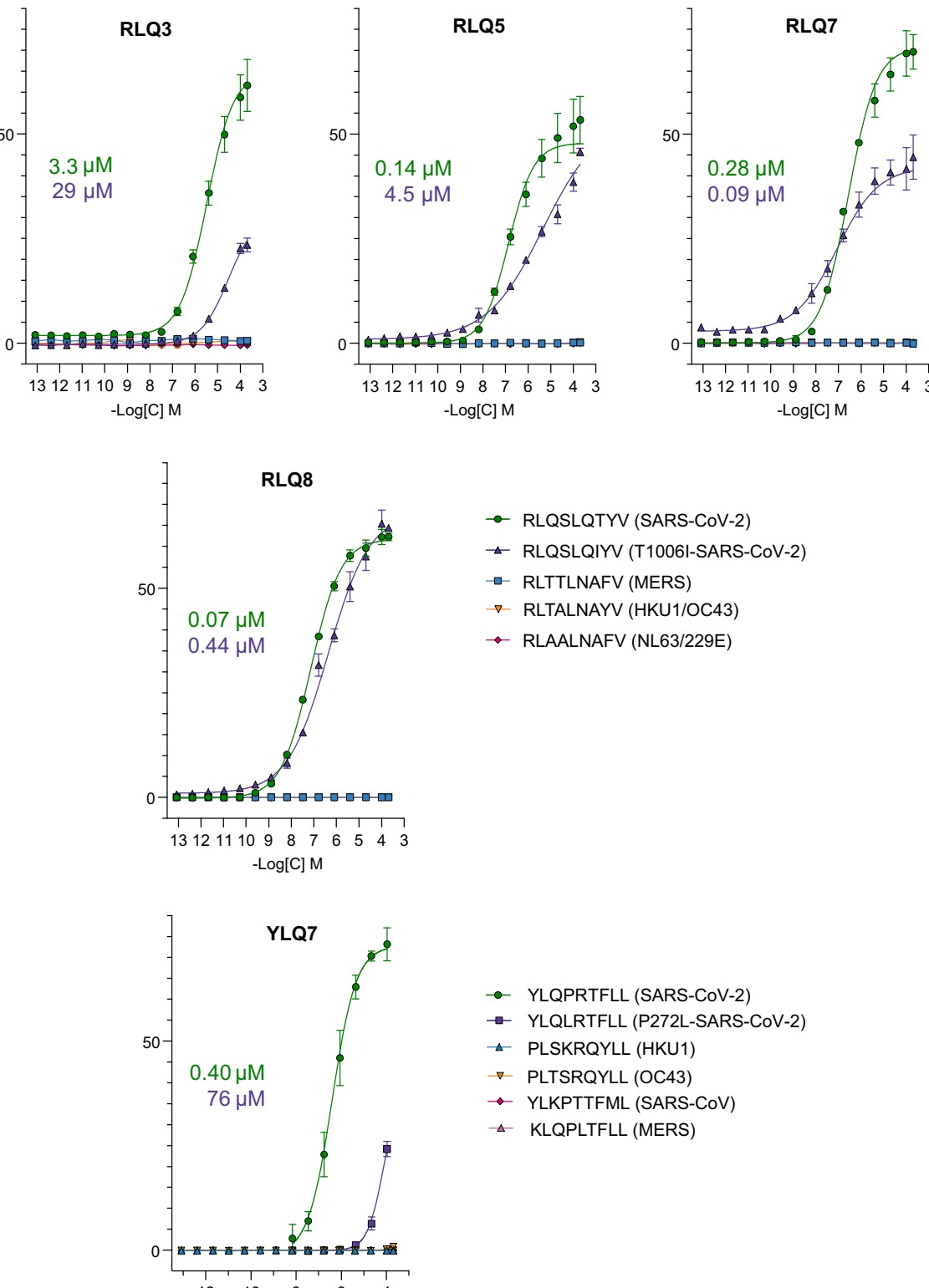

**Fig. 2 T cell activation.** J76 TPR cell line with transgenic TCR were co-cultivated with K562-A*02 cell line loaded with various concentrations of the cognate peptide, mutant peptide, or the homologous peptides from the endemic coronaviruses (n = 3 independent replicates). T cell activation was measured by eGFP expression regulated by the NFAT promoter. Plotted are the mean share of eGFP$^+$ cells and SD. Studied receptor is indicated above each graph. IC$_{50}$ values are shown for the cognate peptide (green) and mutant peptide (violet).

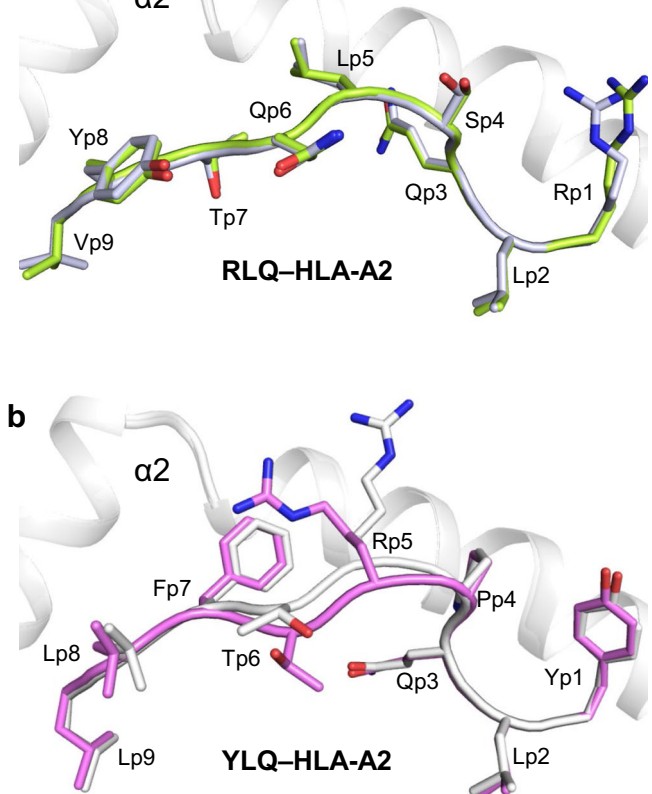

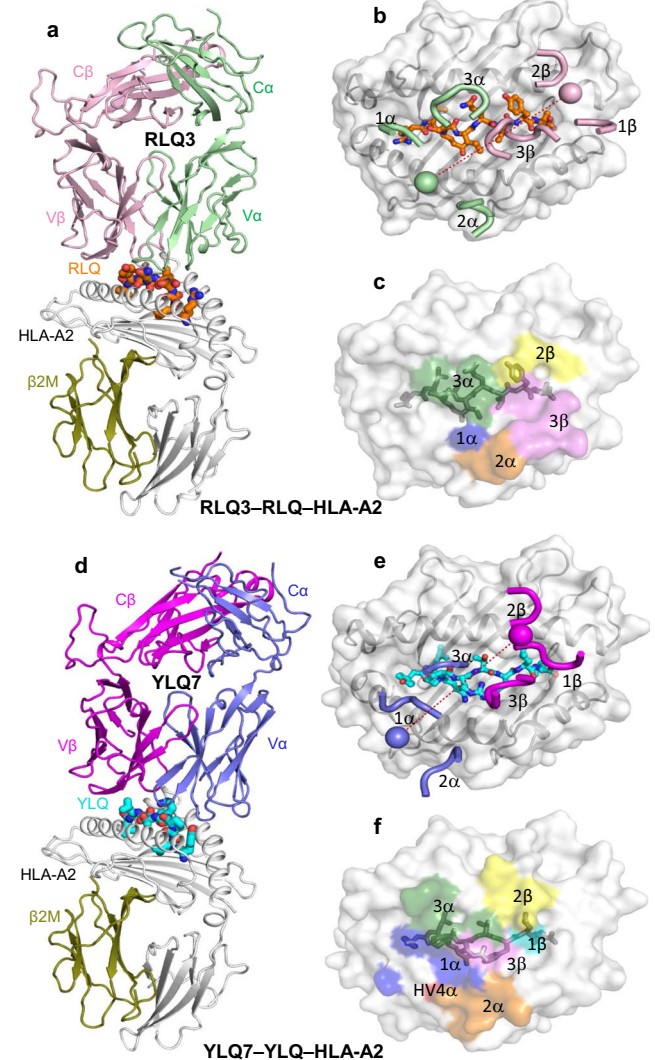

**Fig. 3 Conformations of RLQ and YLQ peptides bound to HLA-A2. a** Side view of two superposed RLQ–HLA-A2 molecules in the asymmetric unit of the crystal. Carbon atoms of the superposed RLQ peptides are light green or gray; nitrogen atoms are blue; oxygen atoms are red. HLA-A2 is gray. Residue labels for RLQ are aligned with the α-carbon atom of the respective residue. **b** Side view of two superposed YLQ–HLA-A2 molecules in the asymmetric unit of the crystal. Carbon atoms of the superposed YLQ peptides are violet or gray.

**Fig. 4 Structure of RLQ3–RLQ–HLA-A2 and YLQ7–YLQ–HLA-A2 complexes. a** Side view of RLQ3–RLQ–HLA-A2 complex (ribbon diagram). TCR α chain, green; TCR β chain, pink; HLA-A2 heavy chain, gray; $β_2$-microglobulin ($β_2$m), olive. The RLQ peptide is orange. **b** Positions of CDR loops of TCR RLQ3 on RLQ–HLA-A2 (top view). CDRs of RLQ3 are shown as numbered green (CDR1α, CDR2α, and CDR3α) or pink (CDR1β, CDR2β, and CDR3β) loops. HLA-A2 is depicted as a gray surface. The RLQ peptide is drawn in orange in stick representation. The green and pink spheres mark the positions of the conserved intrachain disulfide of the Vα and Vβ domains, respectively. The red dashed line indicates the crossing angle of TCR to pMHC. **c** Footprint of TCR RLQ3 on RLQ–HLA-A2. The top of the MHC molecule is depicted as a gray surface. The areas contacted by individual CDR loops are color-coded: CDR1α, blue; CDR2α, orange; CDR3α, green; HV4α, red; CDR1β, cyan; CDR2β, yellow; CDR3β, pink. **d** Side view of YLQ7–YLQ–HLA-A2 complex. TCR α chain, blue; TCR β chain, magenta. The YLQ peptide is cyan. **e** Positions of CDR loops of TCR YLQ7 on YLQ–HLA-A2 (top view). **f** Footprint of TCR YLQ7 on YLQ–HLA-A2.

and CDR3β loops for MHC recognition to approximately the same extent as the germline-encoded CDR1 and CDR2 loops (26 versus 29 contacts).

TCR RLQ3 makes only a few interactions with the HLA-A2 α1 helix (Fig. 5a), mainly through CDR3α Asn92 and CDR2β Asn49 (Supplementary Table 5), as a consequence of the moderately tilted binding mode of RLQ3, which is characterized by an 18° incident angle of TCR over MHC (see above). By contrast, RLQ3 interacts extensively with the HLA-A2 α2 helix via CDR1α, CDR2α, CDR3α, and CDR3β (Fig. 5b), with Vα contributing many more contacts than Vβ, as well as four of five hydrogen bonds: RLQ3 Glu31α Oε2–Nε2 Gln155 HLA-A2, RLQ3 Arg48α Nη1– Nε2 Gln155 HLA-A2, RLQ3 Arg48α Nη2–O Ala150 HLA-A2, and RLQ3 Ser51α Oγ–Nε2 His151 HLA-A2 (Supplementary Table 5). These direct hydrogen bonds are reinforced by six water-mediated hydrogen bonds that further anchor Vα to helix α2 (Fig. 5b).

**RLQ epitope recognition by TCR RLQ3**. In the unliganded RLQ–HLA-A2 structure (Fig. 3a), the RLQ epitope is not very prominent, which is reflected in the relatively small amount of peptide solvent-accessible surface (322 Å$^2$) that TCR RLQ3 buries

upon binding RLQ–HLA-A2. Except for a few interactions involving CDR1α and CDR2β, most contacts between RLQ3 and the RLQ peptide (63 of 79; 80%) are mediated by long CDR3 loops, with CDR3α and CDR3β accounting for 42 and 21 contacts, respectively (Tables 1, 2). TCR RLQ3 engages five residues in the central (P4 Ser, P5 Leu, P6 Gln) and C-terminal terminal (P7 Thr, P8 Tyr) portions of the peptide, but makes no interactions with the N-terminal portion (Fig. 6a) (Supplementary Table 6). The

**Table 1 TCR CDR atomic contacts with peptide and MHC (number of contacts).**

| | α chain | | | | β chain | | | | Total[a] |
|---|---|---|---|---|---|---|---|---|---|
| | CDR1 | CDR2 | HV4 | CDR3 | CDR1 | CDR2 | HV4 | CDR3 | |
| RLQ3 | | | | | | | | | |
| peptide | 4 | 0 | 0 | 42 | 0 | 12 | 0 | 21 | 79 |
| MHC | 4 | 18 | 0 | 6 | 1 | 6 | 0 | 20 | 55 |
| YLQ7 | | | | | | | | | |
| peptide | 15 | 0 | 0 | 28 | 9 | 3 | 0 | 22 | 77 |
| MHC | 13 | 26 | 1 | 6 | 1 | 6 | 0 | 1 | 54 |

Contacts were calculated between non-hydrogen atoms with a 4.0 Å distance cutoff.
[a]Total contacts reflect the total number of TCR–MHC or TCR–peptide contacts.

**Table 2 TCR CDR atomic contacts with peptide and MHC (percentage of contacts).**

| | α chain | | | | β chain | | | |
|---|---|---|---|---|---|---|---|---|
| | CDR1 | CDR2 | HV4 | CDR3 | CDR1 | CDR2 | HV4 | CDR3 |
| RLQ3 | | | | | | | | |
| peptide | 5 | 0 | 0 | 53 | 0 | 15 | 0 | 27 |
| MHC | 7 | 33 | 0 | 11 | 2 | 11 | 0 | 36 |
| YLQ7 | | | | | | | | |
| peptide | 19 | 0 | 0 | 36 | 12 | 4 | 0 | 29 |
| MHC | 24 | 48 | 2 | 11 | 2 | 11 | 0 | 2 |

Contacts were calculated between non-hydrogen atoms with a 4.0 Å distance cutoff.

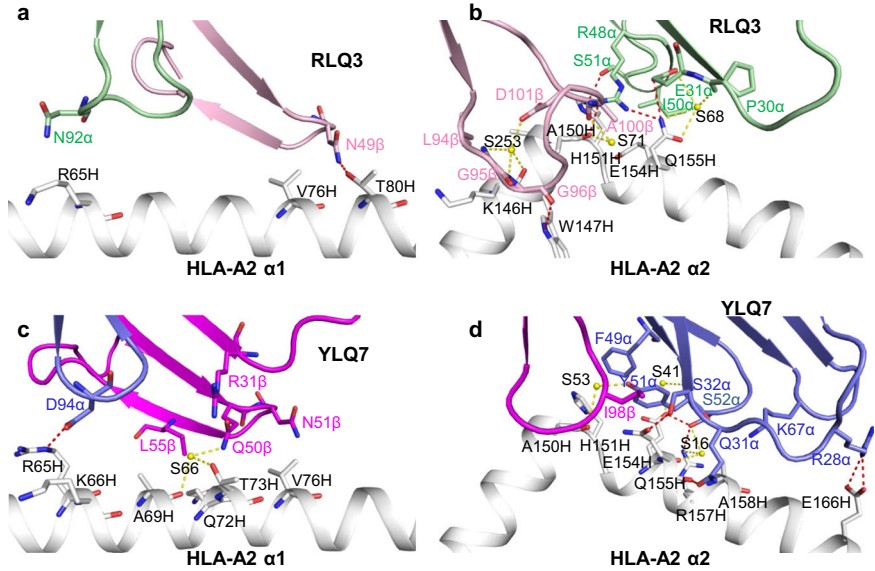

**Fig. 5 Interactions of TCRs with HLA-A2. a** Interactions between RLQ3 and the HLA-A2 α1 helix. The side chains of contacting residues are drawn in stick representation with carbon atoms in green (TCR α chain), pink (TCR β chain) or gray (HLA-A2), nitrogen atoms in blue, and oxygen atoms in red. Hydrogen bonds are indicated by red dashed lines, water molecules are shown as yellow spheres, and water-mediated hydrogen bonds are indicated by yellow dashed lines. **b** Interactions between RLQ3 and the HLA-A2 α2 helix. **c** Interactions between YLQ7 and the HLA-A2 α1 helix. TCR α chain, blue; TCR β chain, magenta. **d** Interactions between YLQ7 and the HLA-A2 α2 helix.

CDR3β loop fits snugly in a notch between the C-terminus of RLQ and the N-terminus of the HLA-A2 α2 helix. The principal focus is on P6 Gln, which alone contributes 22 of 55 van der Waals contacts and 7 of 10 hydrogen bonds with TCR (Fig. 6c). The side chain of P6 Gln inserts into a pocket formed by CDR3α residues Phe91, Gln93, Gly95, and Gln96 (Fig. 6e). Also important for recognition is P8 Tyr, whose side chain packs tightly against that of CDR2β Gln48, and whose main chain forms two hydrogen bonds with CDR3β Gly96 (Fig. 6a). Computational alanine

scanning in Rosetta[50] with the RLQ3–RLQ–HLA-A2 complex structure (Supplementary Table 7) indicates that P6 Gln and P8 Tyr indeed dominate the energetics of the interaction with TCR RLQ3, followed by P5 Leu.

The RLQ3–RLQ–HLA-A2 structure provides a framework for understanding the effects of viral variants and homologous epitopes from other coronaviruses on TCR recognition. We assembled a set of representative spike sequences from 25 human and zoonotic betacoronaviruses and 2 human alphacoronaviruses,

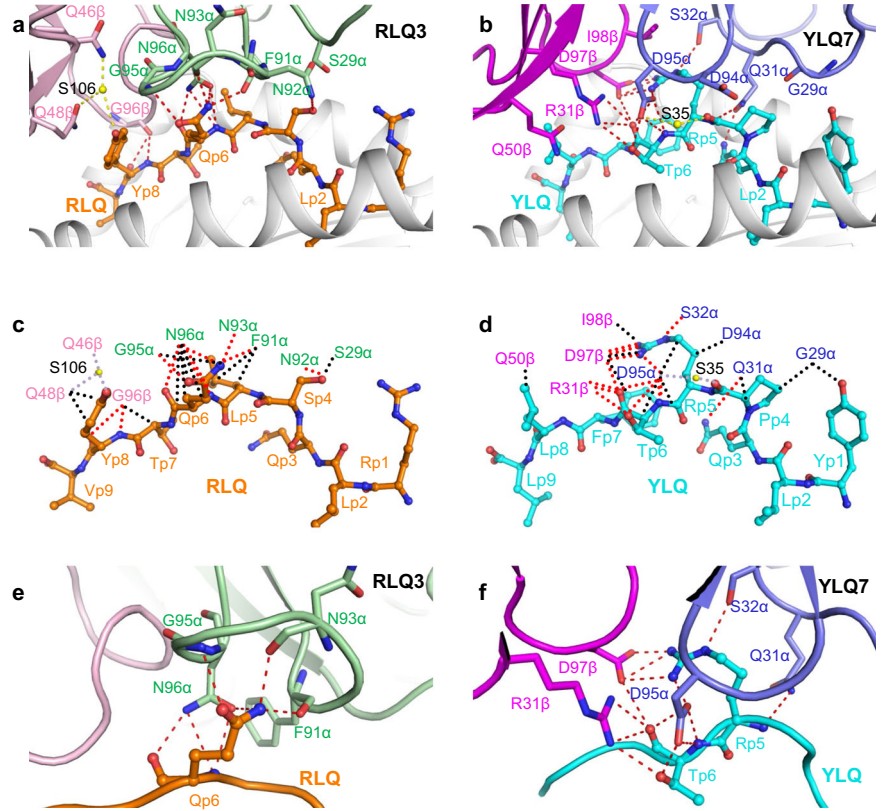

**Fig. 6 Interactions of SARS-CoV-2-specific TCRs with the RLQ and YLQ peptides. a** Interactions between TCR RLQ3 and the RLQ peptide. The side chains of contacting residues are shown in stick representation with carbon atoms in green (TCR α chain), pink (TCR β chain), or orange (RLQ), nitrogen atoms in blue, oxygen atoms in red, and water molecules as yellow spheres. Peptide residues are identified by one-letter amino acid designation followed by position (p) number. Hydrogen bonds are indicated by red dashed lines. Water-mediated hydrogen bonds are drawn as yellow dashed lines. **b** Schematic representation of RLQ3–RLQ interactions. Hydrogen bonds are red dotted lines, water-mediated hydrogen bonds are yellow dotted lines, and van der Waals contacts are black dotted lines. For clarity, not all van der Waals contacts are shown. **c** Close-up of interactions of RLQ3 with P6 Gln of the RLQ peptide. **d** Interactions between TCR YLQ7 and the YLQ peptide (cyan). TCR α chain, blue; TCR β chain, magenta. **e** Schematic representation of YLQ7–YLQ interactions. **f** Close-up of interactions of YLQ7 with P5 Arg and P6 Thr of the YLQ peptide.

and obtained the peptide sequences corresponding to the RLQ epitope site (Supplementary Table 8). Of note, all 12 human and zoonotic betacoronavirus lineage B (SARS-like coronavirus, or sarbecovirus) sequences are fully conserved at the RLQ epitope site, whereas outside of that lineage, most alphacoronavirus and betacoronavirus sequences differ at 4–5 positions within the epitope site. We used computational mutagenesis[50,51] to estimate effects on TCR RLQ3 binding ($\Delta\Delta G$) based on the RLQ–RLQ–HLA-A2 structure, and also used neural network-based predictions of peptide–HLA-A2 affinities[52], to predict effects of RLQ epitope changes on T cell recognition (Supplementary Table 8). As expected due to the sequence differences and our binding measurements for a subset of the epitopes (Fig. 1c–e), generally disruptive effects on RLQ3 TCR binding were predicted. The two SARS-CoV-2 variants of the RLQ epitope that were noted previously (T1006I, Q1005H; Supplementary Table 3) were also modeled, and Q1005H was predicted to result in RLQ3 affinity loss based on the RLQ3–RLQ–HLA-A2 structure, while the affinity disruption of the T1006I epitope was underestimated (prediction of 0.1 Rosetta Energy Units $\Delta\Delta G$, versus 1.0 kcal/mol $\Delta\Delta G$ from our measurements in Fig. 1b).

**Interaction of TCR YLQ7 with HLA-A2.** Of the total contacts between public TCR YLQ7 and HLA-A2 (53), excluding the YLQ peptide, CDR1α, CDR2α, and CDR3α account for 24%, 48%, and 11%, respectively, compared with 2%, 11%, and 2% for CDR1β, CDR2β, and CDR3β, respectively (Table 2). Hence, Vα dominates

the interactions of YLQ7 with MHC (46 of 54 contacts; 85%), with CDR2α contributing far more to the binding interface than any other CDR. In comparison with 154 other TCR–pMHC complexes in the PDB, the YLQ7–YLQ–HLA-A2 complex is the 20th-highest (87th percentile) for the number of atomic CDR2α–pMHC contacts (4 Å distance cutoff).

In sharp contrast to TCR RLQ3, which relies heavily on CDR3α and CDR3β for MHC recognition (see above), nearly all interactions between TCR YLQ7 and MHC are germline-encoded. Thus, YLQ7 contacts the HLA-A2 α2 helix mainly through CDR1α and CDR2α, with Gln31α, Ser32α, and Ser52α forming a dense network of four direct and two water-mediated hydrogen bonds linking YLQ7 to the central section of helix α2 via Glu154H, Gln155H, and Arg157H (Supplementary Table 9) (Fig. 5d). In addition, Arg28α establishes two side-chain–side-chain hydrogen bonds with Glu166H at the C-terminus of helix α2 that provide further stabilization. Similar to RLQ3, YLQ7 makes only sparse contacts with the HLA-A2 α1 helix, primarily via CDR2β (Fig. 5c).

**YLQ epitope recognition by TCR YLQ7.** Unlike TCR RLQ3, which only recognizes the central and C-terminal portions of the RLQ peptide (Fig. 6c), YLQ7 engages all seven solvent-exposed residues along the entire length of YLQ, thereby burying 333 Å$^2$ of peptide surface and enabling maximum readout of the peptide sequence (Supplementary Table 10) (Fig. 6b, d). However, the bulk of interactions between TCR YLQ7 and YLQ involves

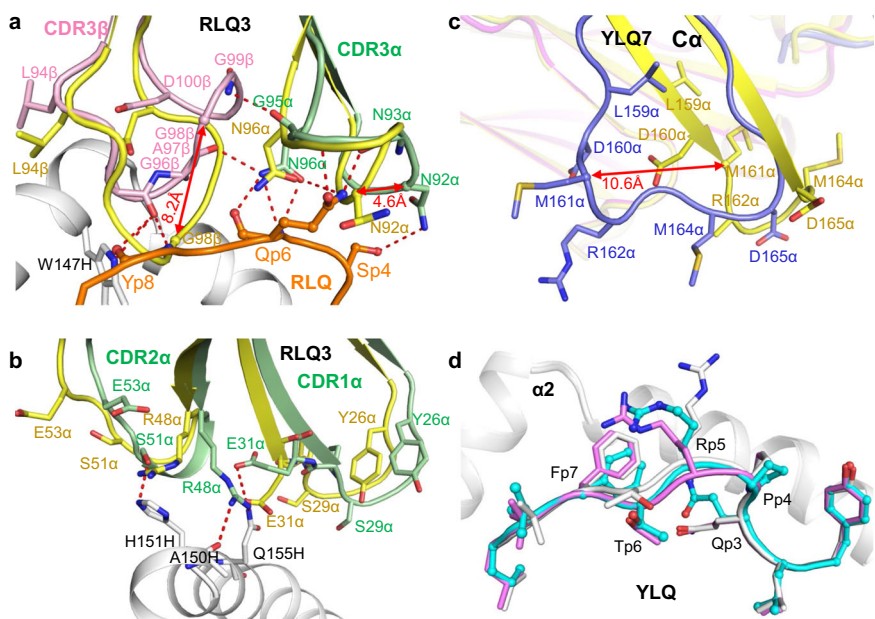

**Fig. 7 Conformational changes in TCRs and pMHC after complex formation. a** Structural rearrangements in CDR3β and CDR3α of RLQ3 (unbound pRLQ3, yellow; bound RLQ3, light green or pink) induced by binding to RLQ–HLA-A2 (RLQ, orange; HLA-A2, gray). Hydrogen bonds are red dotted lines. Double-headed red arrow indicates region of structural shifts. **b** Structural rearrangements in CDR1α and CDR2α of RLQ3. **c** Superposition of the Cα domain of TCR YLQ7 in unbound form and in complex with YLQ–HLA-A2 (unbound YLQ7, yellow; bound YLQ7, blue). Double-headed red arrow indicates site of structural differences in Cα associated with YLQ–HLA-A2 binding. **d** Superposition of YLQ–HLA-A2 in unbound form and in complex with TCR YLQ7 showing rearrangements in residues P4–P7 of the YLQ peptide induced by YLQ7 binding (unbound YLQ, gray or violet; bound YLQ, cyan; HLA-A2, gray).

central residues P5 Arg and P6 Thr: 38 of 62 van der Waals contacts and 14 of 15 hydrogen bonds. These interactions are about evenly distributed between P5 Arg and P6 Thr, which suggests the functional importance of both residues for TCR binding. Computational alanine mutagenesis in Rosetta[50] indicates that both peptide residues are energetically important for TCR YLQ7 binding, with ΔΔGs of 3.0 Rosetta Energy Units (REU, corresponding to energy in kcal/mol) (P5 Arg to Ala) and 1.2 REU (P6 Thr to Ala), and suggests that P5 Arg provides a greater relative contribution to TCR YLQ7 recognition (Supplementary Table 7). Of the 77 total contacts that YLQ7 establishes with the YLQ peptide, CDR1α, CDR2α, and CDR3α account for 19%, 0%, and 36%, respectively, compared with 12%, 4%, and 29% for CDR1β, CDR2β, and CDR3β, respectively (Table 2). Hence, the somatically-generated CDR3 loops dominate TCR interactions with YLQ, with CDR3α and CDR3β making similar overall contributions.

The public CDR3α and CDR3β motifs utilized by YLQ7 may be understood in terms of the YLQ7–YLQ–HLA-A2 structure. The CDR3α motif (89C[AV]VNXXDK[IL]IF99, where X is variable)[26] contains an invariant Asp95α at the tip of the CDR3α loop. This key residue makes extensive interactions (five hydrogen bonds and 14 van der Waals contacts) with P5 Arg and P6 Thr of the YLQ epitope, which are the primary target of YLQ7 (Supplementary Table 10) (Fig. 6f). Similarly, the CDR3β motif (92CASSXDIE[AQ][FY]F102)[26] includes an invariant Asp97β at the tip of the CDR3β loop that, like Asp95α, interacts extensively with P5 Arg and P6 Thr (four hydrogen bonds and six van der Waals contacts). Thus, the need to maintain key interactions with YLQ can explain the selection of conserved CDR3α and CDR3β sequences in TCRs from different individuals.

The YLQ7–YLQ–HLA-A2 structure also provides insights into the selection of particular TRAV and TRBV gene segments. The large majority (85%) of HLA-A*0201-restricted, YLQ-specific

TCRs from COVID-19 CPs were found to utilize TRAV12-1 or TRAV12-2; none used the nearly identical TRAV12-3 gene segment[26]. Both TRAV12-1 and TRAV12-2 encode CDR1α residues Gln31 and Ser32, whereas TRAV12-3 encodes CDR1α residues Gln31 and Tyr32 (Supplementary Fig. 6). Substitution of Ser32α with Tyr32α is predicted to disrupt YLQ7 binding to YLQ–HLA-A2, based on computational mutagenesis in Rosetta (ΔΔG: 3.7 REU), due in part to the loss of the hydrogen bonding interaction between Ser32α and Gln155 of the MHC. The TRBV gene segment most frequently used by YLQ-specific TCRs, including YLQ7, is TRBV7-9[26]. Other members of the TRBV7 family occurred much less frequently (TRBV7-2 and TRBV7-8) or not at all (TRBV7-1 and TRBV7-3 through TRBV7-7). One unique feature of TRBV7-9 is an arginine at position 31 (Supplementary Fig. 6); in YLQ7, the Arg31β side chain forms part of a network of polar interactions with the YLQ peptide (Fig. 6f). In other TRBV7 germline genes, the residue at this position is serine, threonine, or alanine, none of which would be capable of mediating these critical polar interactions, thus providing a possible mechanistic explanation for the TRBV7-9 gene preference.

The YLQ7–YLQ–HLA-A2 structure enables the prediction and mechanistic understanding of effects of viral variants and homologous epitopes from other coronaviruses on TCR recognition or MHC presentation. YLQ epitope orthologous sequences were identified from the same set of 25 representative human and zoonotic coronaviruses used in analysis of RLQ epitope orthologs (Supplementary Table 8). The YLQ epitope exhibits considerably more variability across coronaviruses than the RLQ epitope, ranging from fully conserved for the bat coronavirus RaTG13, to substitutions at nearly every position in other betacoronaviruses and alphacoronaviruses. In contrast with the RLQ sequence, the YLQ sequence varies within the sarbecovirus lineage (lineage B), with as many as six substitutions from the SARS-CoV-2 sequence (RmYN02). YLQ7 TCR binding, and/or HLA-A2 MHC binding,

was predicted to be disrupted for all of the peptide orthologs not matching the SARS-CoV-2 YLQ sequence. One pangolin sarbecovirus peptide sequence from this set (GD_pangolin; Genbank ID QIG55945.1), with predicted maintained HLA-A2 binding and predicted loss of YLQ7 binding, was tested experimentally and confirmed to lead to a marked reduction of YLQ7 TCR binding affinity ($\Delta\Delta G = 1.7$ kcal/mol). YLQ peptide SARS-CoV-2 variants were also assessed for predicted HLA-A2 presentation and TCR YLQ7 binding (Supplementary Table 8). L270F, an MHC anchor residue substitution which was reported by others due to its capacity for HLA-A2 binding disruption[17], as expected was found to have high predicted HLA-A2 affinity loss (as confirmed experimentally[17]). Most other YLQ peptide variants were predicted to be destabilizing for TCR YLQ7, with the exception of substitutions at position P272, including P272L, which were predicted to have neutral or minor stabilizing effects on YLQ7 binding.

**Conformational changes in TCRs and upon pMHC binding.** To assess ligand-induced conformational changes in the TCRs, we determined the structures of RLQ3 and YLQ7 in the unbound form to 1.88 and 2.35 Å resolution, respectively (Supplementary Table 4). Superposition of the VαVβ domains of free RLQ3 onto those in complex with RLQ–HLA-A2 revealed structural differences in CDR1α, CDR2α, CDR3α, and CDR3β. The CDR3β loop underwent a large movement (r.m.s.d. in α-carbon positions of 3.0 Å for residues 94–101), thereby enabling CDR3β to insert into a notch between the C-terminus of RLQ and the N-terminus of the HLA-A2 α2 helix (Fig. 7a). CDR3β Gly98 showed the largest individual displacement (8.2 Å in its α-carbon position). CDR3α underwent a rearrangement (r.m.s.d. in α-carbon positions of 2.0 Å for residues 90–97) that resulted in formation of eight hydrogen bonds and 34 hydrophobic contacts with P4 Ser, P5 Leu, and P6 Gln. CDR3α Asn92 showed the largest individual shift (4.6 Å in its α-carbon position). CDR1α and CDR2α displayed small yet relevant movements (r.m.s.d. in α-carbon positions of 2.7 and 2.0 Å for residues 26–30 and 50–54, respectively) that allow them to engage the HLA-A2 α2 helix and RLQ peptide via four hydrogen bonds and 22 van der Waals contacts (Fig. 7b).

Superposition of the VαVβ domains of unbound YLQ7 onto those in complex with YLQ–HLA-A2 revealed that conformational adjustments in CDR loops were restricted mainly to shifts in side-chain orientation that serve to maximize interactions with pMHC. Surprisingly, the Cα domain of bound YLQ7 showed a large deviation from the Cα domain of unbound YLQ7 (Fig. 7c), as well as from all previously reported Cα structures (Supplementary Fig. 7). Cα residues 157–165, which are in unambiguous electron density in both free and bound TCR structures, adopt markedly different main-chain conformations, with r.m.s.d. in α-carbon positions of 6.0 Å. Cα Met161 showed the largest individual displacement (10.6 Å in its α-carbon position). In bound YLQ7, β-strand D ends prematurely at Cα Leu159 compared to unbound YLQ7. As a result, the β-hairpin formed by strands D and E in a typical Cα domain is disrupted and residues 157–165 assume a loop configuration (Fig. 7c). Whether this structural rearrangement is a consequence of YLQ–HLA-A2 binding or simply reflects a degree of malleability in Cα is unclear (see Discussion). In addition, superposition of the MHC α1α2 domains of free YLQ–HLA-A2 onto those of YLQ–HLA-A2 in complex with YLQ7 showed that TCR binding stabilizes the central portion of the YLQ peptide in a conformation intermediate between the two observed in the unbound YLQ–HLA-A2 structure (Fig. 7d), thus optimizing TCR interactions with both peptide and MHC.

## Discussion

In most T cell responses, the TCR repertoires elicited by a particular antigenic epitope are distinct between individuals (private T cell responses). By contrast, certain other epitope-specific TCR repertoires contain TCRs that are frequently observed in multiple unrelated individuals (public T cell responses). Public TCRs have been described in immune responses to a variety of human viruses, including CMV, HIV, EBV[53], and, more recently, SARS-CoV-2[26]. Thus, the YLQ spike epitope elicited highly public TCRs among COVID-19 CPs, as exemplified by TCR YLQ7. By contrast, TCRs elicited by the RLQ spike epitope, such as TCR RLQ3, were found to be largely private[26], as were TCRs elicited by a SARS-CoV-2 nucleocapsid epitope (B7/N$_{105}$)[32].

A possible limitation of this study is the frequency-based pairing of TCR α and β chains. We therefore cannot formally exclude that some RLQ-specific TCRs might be unnaturally paired, even though they bound RLQ–HLA-A2 with affinities characteristic of TCRs with high functional avidity for microbial antigens ($K_D < 50$ μM)[41]. By contrast, the natural pairing of YLQ7 α and β chains was independently confirmed in another study using single-cell sequencing[32].

TRAV12-1/12-2 and TRBV7-9 were used by 85 and 21% of YLQ-specific TCRs and is the predominant Vα/Vβ combination[26]. The strong bias for these V genes suggests the importance of germline-encoded features in TCR recognition of the YLQ–HLA-A2 ligand. TRAV12-1 and TRAV12-2 both encode CDR1α Ser32, whereas TRAV12-3, which is nearly identical to TRAV12-1/12-2 but is not selected in the YLQ-specific repertoire, encodes CDR1α Tyr32 (Supplementary Fig. 6). The hydroxyl group of the Ser32 side chain in TCR YLQ7 participates in multiple polar contacts, engaging the MHC residue Gln155 and Arg6 of the YLQ peptide, and this polar network would likely not be possible with a bulkier Tyr residue at position 32, thus providing a basis for TRAV12-1/12-2 bias. Similarly, TRBV7-9 encodes CDR1β Arg31, whose side chain forms multiple hydrogen bonds with P6 Thr of the YLQ peptide. Other TRBV7 family members, which are not selected in the YLQ-specific repertoire[26], encode CDR1β Ser31, Thr31 or Ala31 (Supplementary Fig. 6), whose smaller and uncharged side chains cannot replicate these key interactions.

The private nature of TCR RLQ3 may be explained, at least in part, by its heavy reliance on the somatically-generated CDR3α and CDR3β loops for MHC (as well as peptide) recognition, whereas nearly all interactions between YLQ7 and MHC are germline-encoded. This reduces the likelihood of mechanistically forming identical or very similar V(D)J rearrangements in different individuals that are still compatible with pMHC recognition (convergent recombination)[54,55]. Nevertheless, multiple distinct solutions do exist to binding RLQ–HLA-A2, as demonstrated by TCRs RLQ5, RLQ7, and RLQ8, which use α/β chain pairs completely distinct from RLQ3 and from each other.

Several studies have revealed the presence of CD4+ and CD8+ T cells recognizing SARS-CoV-2 epitopes in unexposed individuals[27,30,34–36]. The possibility that pre-existing T cell immunity to SARS-CoV-2 can be induced by seasonal human coronaviruses such as NL63, OC43, and HKU1 is supported by a relatively high amino acid similarity between recognized SARS-CoV-2 epitopes and homologous sequences from these other viruses. However, we did not observe any interaction of TCRs YLQ7, RLQ3, RLQ5, RLQ7, or RLQ8 with homologous epitopes from NL63, OC43, or HKU1, either by SPR or in T cell activation assays. This lack of cross-reactivity is consistent with predictions of TCR–pMHC affinity from computational mutagenesis, which predicted disruption of pMHC binding of TCRs RLQ3 and YLQ7 for peptides from those human coronaviruses, as well as peptides

from zoonotic coronaviruses that contain one or more substitutions in the SARS-CoV-2 epitope sequence.

We found that two natural variants of the YLQ and RLQ epitopes (P272L and T1006I) that contain the most commons epitope mutations in the GISAID database[45] had different effect on TCR recognition. Despite 1.3- to 6.8-fold reductions in TCR affinity as measured by SPR, T1006I activated RLQ3, RLQ5, RLQ7, and RLQ8 T cells in functional assays nearly as efficiently as the wild-type epitope. Since the RLQ response is oligoclonal[26], the efficiency of recognition of the T1006I variant could vary among patients depending on the involvement of particular clones. For TCR YLQ7, the 64-fold reduction of affinity for the P272L variant corresponded to a 189-fold reduction of functional avidity that disabled cellular activation at physiological peptide concentrations. This agrees with a recent study reporting that the P272L variant is unable to activate primary T cell cultures and that this mutation is under natural selection in European populations with high incidence of the HLA-A*02:01 allele[46]. Another reported variant of the YLQ epitope (L270F) was shown to be non-immunogenic due to decreased stability of the pMHC complex[17]. In the YLQ–HLA-A2 structure, Leu270 serves as the P2 anchor residue and occupies pocket B of HLA-A2.

Although SARS-CoV-2 evolution is more obvious in the accumulation of mutations that increase infectivity or evade neutralizing antibodies[56], some level of T cell evasion is also detectable. Thus, mutations in several HLA class I-restricted SARS-CoV-2 epitopes besides YLQ were found to potentially enable the virus to escape killing by cytotoxic CD8+ T cells[17] and it is predicted that emerging variants of concern such as the Alpha, Beta, and Delta variants have a substantial number of peptides with decreased binding to common HLA class I alleles[57]. However, as T cell responses to SARS-CoV-2 are targeted to multiple epitopes simultaneously[6,10,13,17,24–33], it is not expected that any single mutation can radically influence the overall magnitude of the response, an important consideration in vaccine development. Indeed, a recent study found that most immunogenic epitopes were conserved in several of the emerging variants and that there was no detectable decrease of T cell reactivity to these strains in either vaccinated or convalescent patients[58].

The Cα domain of bound TCR YLQ7 exhibited a main-chain conformation remarkably different from those previously reported for TCR structures. Whereas free YLQ7 has a typical Cα structure (Supplementary Fig. 7a), Cα β-strand D in bound YLQ7 terminates prematurely at Leu159, causing residues 157–165 to shift from a β-hairpin to a loop conformation and leading to significant changes in Cα topology (Supplementary Fig. 7b). Intriguingly, one other TCR structure (1F1E8hu) also showed an atypical Cα structure characterized by β-strand slippage of residues 157–170 (Supplementary Fig. 7c) that was proposed to represent a signaling intermediate[59]. Although the atypical Cα conformations of TCRs YLQ7 and 1F1E8hu are clearly different, they involve the same region of Cα, pinpointing a site of structural plasticity. In the cryoEM structure of the TCR–CD3 complex (Supplementary Fig. 7d)[60], Cα residues Arg162, Ser163, and Asp165 at the tip of the β-hairpin formed by strands D and E contact the CD3δ subunit of the CD3εδ heterodimer (Supplementary Fig. 7e). These interactions are incompatible with the atypical Cα conformations of YLQ7 or 1F1E8hu. As such, their disruption could alter the quaternary structure of the TCR–CD3 complex and thereby affect T cell signaling, a hypothesis that warrants further investigation.

While currently available vaccines against SARS-CoV-2 are effective against that virus[61,62], albeit with reduced immunized serum antibody neutralization against some variants[56,63], a major unmet need is pan-coronavirus vaccine candidates that can protect against infection from prospective emergent coronaviruses, in addition to SARS-CoV-2 and its variants. Such efforts can be informed by recent studies that have described conserved antibody epitopes on the spike glycoprotein and the structural basis of their targeting by cross-reactive monoclonal antibodies;[64–66] such cryptic and sub-dominant epitopes can be the target of efforts to engineer antigens to focus the antibody response to these epitopes. An additional consideration in pan-coronavirus vaccine design is the effective induction of T cell responses to epitopes that are conserved across coronaviruses, such as the RLQ epitope. This point is underscored by this study, where through structural determination, binding experiments, and computational analysis, we have highlighted the exquisite specificity of human TCRs that target two T cell epitopes from SARS-CoV-2. These TCRs recognize sites in the N-terminal domain (NTD; YLQ epitope) and central helix (CH; RLQ epitope) regions of the spike glycoprotein that are partially or fully buried in the spike, and in the case of the RLQ epitope, represent a conserved site of vulnerability that is inaccessible to antibodies yet targeted by T cells. By delineating the mechanistic basis of TCR targeting of an immunodominant yet variable site, and a conserved and less commonly targeted site, this study provides useful information for prospective efforts to rationally design and optimize effective vaccines that are capable of long-lasting and cross-protective immunity against coronaviruses. In a recent example of structure-guided T cell vaccine design, structure-based network analysis was used to identify mutationally constrained CD8+ T cell epitopes that are conserved across SARS-CoV-2 variants and sarbecoviruses[67].

## Methods

**Peptide titration assay.** Triple reporter J76 cells[42] were transduced with lentiviral vectors encoding the α and β chains of selected TCR separated by p2A peptide sequences and expressed under control of the EF1 promoter. Except where it is indicated otherwise, all lentiviral constructs also contained CD8α and CD8β separated from TCR by t2A and from each other by p2A peptide sequences. $1.25 \times 10^5$ TCR transgenic J76 cells were co-incubated with $2.5 \times 10^5$ K562 cells transgenic for HLA-A*02:01 in 96 well plates with 200 μl of IMDM media 10% FCS (Gibco) containing serial dilutions of peptide in three independent replicates. In CD8 blocking experiments, cells were preincubated with 7 μg/ml CD8 blocking antibody clone SK-1 (BD Biosciences) which was also present throughout the stimulation at a concentration of 1.75 μg/ml. Media without peptide was used as a negative control. After 16 h of incubation at 37 °C in 5% $CO_2$, cells were washed with PBS and surface stained with CD8-APC (BD Biosciences). Cell viability was assessed by staining with Alexa Fluor 750 NHS Ester (ThermoFisher) according to the manufacturer's recommendations. T cell activation by peptide was assessed according to the expression of eGFP regulated by the NFAT promoter and analyzed on a MACSQuant Analyzer 10 (Miltenyi Biotec). The acquired data were processed by FlowJo (version 10.6.2) and Prizm Software for analysis. Percent of eGFP expression cells was calculated in the CD8+ or CD3+ gate. For gating strategy see Supplementary Fig. 1c. Negative control values were subtracted.

**Protein preparation.** The sequencing of RLQ- and YLQ-specific TCRs from COVID-19 CPs was described previously[26]. Soluble TCRs RLQ3 and YLQ7 for affinity measurements and structure determinations were produced by in vitro folding from inclusion bodies expressed in *Escherichia coli*. Codon-optimized genes encoding the α and β chains of these TCRs (TCR RLQ3 residues 1–204 and 1–244; TCR YLQ7 residues 1–203 and 1–241, respectively) were synthesized (Supplementary Table 11) and cloned into the expression vector pET22b (GenScript). An interchain disulfide (CαCys158–CβCys171 in RLQ3; CαCys157–CβCys168 in YLQ7) was engineered to increase the folding yield of TCR αβ heterodimers. The mutated α and β chains were expressed separately as inclusion bodies in BL21(DE3) *E. coli* cells (Agilent Technologies). Bacteria were grown at 37 °C in LB medium to $OD_{600} = 0.6$–0.8 and induced with 1 mM isopropyl-β-D-thiogalactoside. After incubation for 3 h, the bacteria were harvested by centrifugation and resuspended in 50 mM Tris-HCl (pH 8.0) containing 0.1 M NaCl and 2 mM EDTA. Cells were disrupted by sonication. Inclusion bodies were washed with 50 mM Tris-HCl (pH 8.0) and 5% (v/v) Triton X-100, then dissolved in 8 M urea, 50 mM Tris-HCl (pH 8.0), 10 mM EDTA, and 10 mM DTT. For in vitro folding, the TCR α (45 mg) and β (35 mg) chains were mixed and diluted into 1 liter folding buffer containing 5 M urea, 0.4 M L-arginine-HCl, 100 mM Tris-HCl (pH 8.0), 3.7 mM cystamine, and 6.6 mM cysteamine. After dialysis against 10 mM Tris-HCl (pH 8.0) for 72 h at 4 °C, the folding mixture was concentrated 20-fold and dialyzed against 50 mM MES buffer (pH 6.0). After removal

of the precipitate formed at pH 6.0 by centrifugation, the supernatant was dialyzed overnight at 4 °C against 20 mM Tris-HCl (pH 8.0), 20 mM NaCl. Disulfide-linked RLQ3 and YLQ7 TCR heterodimers were purified using consecutive Superdex 200 (20 mM Tris-HCl (pH 8.0), 20 mM NaCl) and Mono Q (20 mM Tris-HCl (pH 8.0), 0–1.0 M NaCl gradient) FPLC columns (GE Healthcare).

Soluble HLA-A2 loaded with RLQ peptide (RLQSLQTYV), YLQ (YLQPRTFLL) peptide, or other peptides (Supplementary Table 2) was prepared by in vitro folding of E. coli inclusion bodies as described[68]. Correctly folded RLQ–HLA-A2, YLQ–HLA-A2, and other peptide–HLA-A2 complexes were purified using sequential Superdex 200 (20 mM Tris-HCl (pH 8.0), 20 mM NaCl) and Mono Q columns (20 mM Tris-HCl (pH 8.0), 0–1.0 M NaCl gradient). To produce biotinylated HLA-A2, a C-terminal tag (GGGLNDIFEAQKIEWHE) was attached to the HLA-A*0201 heavy chain. Biotinylation was carried out with BirA biotin ligase (Avidity).

**Crystallization and data collection**. For crystallization of TCR–pMHC complexes, TCRs RLQ3 and YLQ7 were mixed with RLQ–HLA-A2 and YLQ–HLA-A2, respectively, in a 1:1 and concentrated to 10 mg/ml. Crystals were obtained at room temperature by vapor diffusion in hanging drops. The RLQ3–RLQ–HLA-A2 complex crystallized in 0.2 M ammonium sulfate, 0.1 M MES (pH 6.0), and 12% (w/v) polyethylene glycol (PEG) 4000. Crystals of the YLQ7–YLQ–HLA-A2 complex grew in 0.1 M ammonium sulfate, 0.3 M sodium formate, 0.1 M sodium acetate (pH 5.0), 3% (w/v) γ-polyglutamic acid (Na$^+$ form, LM), and 3% (w/v) PEG 20000. Crystals of RLQ–HLA-A2 were obtained in 0.2 M ammonium sulfate, 0.1 M MES (pH 6.5), and 20% (w/v) PEG 8000 by micro-seeding. Crystals of YLQ–HLA-A2 grew in 0.2 M potassium thiocyanate (pH 7.0) and 22% (w/v) PEG 3350. Crystals of unbound TCR RLQ3 were obtained in 0.2 M calcium acetate, 0.1 M imidazole (pH 8.0), and 17% (w/v) PEG 1500. Unbound TCR YLQ7 crystallized in 1.2 M potassium sodium tartrate tetrahydrate and 0.1 M Tris-HCl (pH 8.0). Before data collection, all crystals were cryoprotected with 20% (w/v) glycerol and flash-cooled. X-ray diffraction data were collected at beamline 23-ID-D of the Advanced Photon Source, Argonne National Laboratory. Diffraction data were indexed, integrated, and scaled using the program HKL-3000[69]. Data collection statistics are shown in Supplementary Table 4.

**Structure determination and refinement**. Before structure determination and refinement, all data reductions were performed using the CCP4 software suite[70]. Structures were determined by molecular replacement with the program Phaser[71] and refined with Phenix[72]. The models were further refined by manual model building with Coot[73] based on $2F_o – F_c$ and $F_o – F_c$ maps. The α chain of TCR 42F3 (PDB accession code 3TFK)[74], the β chain of anti-EBV TCR CF34 (3FFC)[75], and p53R175H–HLA-A2 (6VR5)[68] with the CDRs and peptide removed were used as search models to determine the orientation and position of the RLQ3–RLQ–HLA-A2 complex. The orientation and position parameters of unbound TCR RLQ3 and RLQ–HLA-A2 were obtained using the corresponding components of the RLQ3–RLQ–HLA-A2 complex. Similarly, the α chain of riboflavin-specific TCR D462-E4 (6XQP)[76], the β chain of a staphylococcal enterotoxin E-bound TCR (4UDT)[77], and p53R175H–HLA-A2 (6VR5)[68] with the CDRs and peptide removed were used as search models for molecular replacement to determine the structure of the YLQ7–YLQ–HLA-A2 complex. The corresponding components of the YLQ7–YLQ–HLA-A2 complex were used as search models to determine the coordinates of unbound YLQ7 and YLQ–HLA-A2. Refinement statistics are summarized in Supplementary Table 4. Contact residues were identified with the CONTACT program[70] and were defined as residues containing an atom 4.0 Å or less from a residue of the binding partner. The PyMOL program (https://pymol.org/) was used to prepare figures.

**Surface plasmon resonance analysis**. The interaction of TCRs RLQ3 and YLQ7 with pMHC was assessed by surface plasmon resonance (SPR) using a BIACore T100 biosensor at 25 °C. Biotinylated RLQ–HLA-A2, YLQ–HLA-A2, or other peptides–HLA-A2 ligand was immobilized on a streptavidin-coated BIAcore SA chip (GE Healthcare) at around 1000 resonance units (RU). The remaining streptavidin sites were blocked with 20 μM biotin solution. An additional flow cell was injected with free biotin alone to serve as a blank control. For analysis of TCR binding, solutions containing different concentrations of RLQ3 or YLQ7 were flowed sequentially (50 μl/min, 600 s for dissociation) over chips immobilized with RLQ–HLA-A2, YLQ–HLA-A2, other peptides–HLA-A2 ligand, or the blank. Dissociation constants ($K_{DS}$) were calculated by fitting equilibrium and kinetic data to a 1:1 binding model using BIA evaluation 3.1 software.

**Computational sequence and structural analysis**. YLQ and RLQ epitope variants and their frequencies were obtained from the GISAID database (www.gisaid.org)[45] based on the counts of annotated variants within the corresponding SARS-CoV-2 spike glycoprotein sequence ranges. These data were obtained in May 2021, and frequencies are from a total of approximately 1.6 million spike glycoprotein sequences present in the database. Representative spike glycoprotein sequences for other coronaviruses, corresponding to an adaptation of a set of spike sequences from the CoV3D database[78], were obtained from NCBI and GISAID, and aligned using MAFFT software[79] to generate a multiple sequence

alignment which was used to obtain sequences corresponding to the YLQ and RLQ epitope positions in those viruses. Betacoronavirus lineage and clade information was determined based on previously defined classifications of coronaviruses[80,81], and phylogenetic comparison of spike protein sequences as described for CoV3D[78]. Computational prediction of HLA-A2 binding affinities ($IC_{50}$ values) for YLQ and RLQ epitope sequences, and variants thereof, was performed with the NetMHCPan 4.1 algorithm[52], on the Immune Epitope Database (IEDB) tools site. Prediction of RLQ3 and YLQ7 TCR binding effects ($\Delta\Delta Gs$) for epitope variants and orthologs was performed using computational mutagenesis in Rosetta (v.2.3)[50], which was previously used to predict TCR–pMHC affinity changes for other TCRs[51].

**Reporting Summary**. Further information on research design is available in the Nature Research Reporting Summary linked to this article.

## Data availability
Atomic coordinates and structure factors have been deposited in the Protein Data Bank under accession codes 7N1A (YLQ–HLA-A2), 7N1B (RLQ–HLA-A2), 7N1C (RLQ3), 7N1D (YLQ7), 7N1E (RLQ3–RLQ–HLA-A2), and 7N1F (YLQ7–YLQ–HLA-A2).

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

## Acknowledgements

This work was supported by National Institutes of Health Grants GM126299 (to B.G.P.) and AI129893 (to R.A.M.), by Russian Science Foundation Grant 20-15-00395 (to G.A.E.), and by National Natural Science Foundation of China Grant 32100985 (to

D.W.). Results in this report are based on work performed at both Structural Biology Center and GM/CA beamlines at the Advanced Photon Source of Argonne National Laboratory, operated by UChicago Argonne, LLC, for the U.S. Department of Energy, Office of Biological and Environmental Research under contract DE-AC02-06CH11357. Computing resources from the University of Maryland Institute for Bioscience and Biotechnology Research High-Performance Computing Cluster were used in this study. We thank Dongxiu Zhang Spiering for assistance with TCR expression.

## Author contributions

D.W., A.K., R.Y., J.D.G., R.G., A.S., Y.S., and D.V.D. performed the experiments and data analyses. G.A.E., B.G.P., and R.A.M. conceived and supervised the project. All authors prepared the paper.

## Competing interests

The authors declare no competing interests.
