## [Peer Review File · Nature Communications]

Structural assessment of HLA-A2-restricted SARS-CoV-2 spike epitopes recognised by public and private T-cell receptorsREVIEWER COMMENTS

Reviewer #1 (Remarks to the Author):

In the paper titled “Structural basis for recognition of two HLA-A2-restricted SARS-CoV-2 spike epitopes by public and private T cell receptors” Wu and colleagues have used structural, computational, biochemical and cellular tools to characterize the binding properties of a public and a private TCR identified from SARS-CoV-2 infected individuals. They find that the public TCR is more driven by germline contacts between the TCR and MHC, while the private TCR relies more upon CDR loops. They also find a degree of cross-reactivity within each TCR, albeit not any sufficient to confer broad cross-reactivity in these cases. This work is timely, relevant, and will be of interest as they represent some of the first TCR structures bound to SARS-CoV-2 ligands. However, answering a few remaining questions would improve the work as presented.

Major points:

1. The affinity between RLQ3 and RLQ/A2 is reported as mid micromolar, which is weak for alpha beta TCRs affinity for viral pMHCs, and significantly weaker than two of the other RLQ-binding TCRs. Ideally, the authors would show additional evidence that the alpha-beta pairs are correct, not inadvertently mismatched.

a. Towards this goal, showing the TRAV and TRBV frequency data from the patient sample in a supplement could be helpful.

b. Alternatively, if it is available, single cell analysis would validate the pairing.

2. While overall reasonably conducted, there are aspects of the SPR in Figures 1 and S1 that could be improved:

a. The binding kinetics of many of the measured interactions (other than RQL5 to RQL/HLA-A2) are either too fast or approaching too fast to be measured accurately by the instrument ($\sim 0.5\text{s}^{-1}$ for koff). Therefore, the authors should show the residuals fitting of the kinetic parameters. It would also be reasonable to only report equilibrium measurements, as the kinetics do not factor into analysis of the paper.

b. Several of the equilibrium measurements (Fig 1b, 1h, S1c,d,f) do not have a sufficient number of high concentration datapoints to accurately determine an equilibrium K_d (either 0 or 1 points over the calculated K_d , which is insufficient for a high-confidence curve fit). The authors should either re-measure the values, state them as N.D. or as greater than some value.

c. Some of the sensograms for the T1006I measurements do not come to equilibrium, suggesting non-specific binding. If possible, these measurements should be redone.

d. Authors should include the flowrates used for the kinetic experiments and the dissociation time post injection.

3. I personally found the inclusion of the Rosetta-based analyses not valuable without any external validation. This is made clear by the fact that one of the Rosetta measurements with experimental validation (assessing the P272L mutation) does not match the experimental observations. However, these results are also not particularly central to the paper: I would consider de-emphasizing these because they may add more noise than signal to the work, or alternatively further buttress them with further experimental validation. This would be onerous in terms of making proteins, but would be relatively straightforward to functionally test on the reporter cell lines at least in case of the peptide mutations.

Clarity and formatting:

4. I found the paper, at times, confusing to read. This was not a problem on a micro level (each part was well written), but rather in the macro sense: the early paper describes YLQ before RLQ, before switching order at the structures. The paper also describes RLQ in detail before YLQ, essentially requiring the reader to work through figures 3-6 twice. Some restructuring would help the readers out a lot. Along with this, consistent coloring for any given structure between figures, and better labeling of the overall structures within a figure (making it more clear which structure is which in Figures 4-7) would be helpful.

5. Adding virus names to Figure 2 so the reader is not left with just the amino acid sequences would be helpful.

6. Fig 3A: The difference in R_{work} vs R_{free} is 7% for RLQ/A2 structure; additionally, despite the moderate resolution of the structure the R_{work} is 0.201. Could this be evidence of overfitting?

7. I believe the last figure reference before the discussion (pg 18) should reference 7d rather than 6d.
8. The structural basis of public responses is a quite interesting topic – the authors discuss this but do not have it clearly laid out in one figure. Collating a figure that directly shows the basis for these V regions would be interesting (either for main text or supplement).

Reviewer #2 (Remarks to the Author):

The manuscript by Wu et al examines the first structures of TCR complexes containing SARS-CoV-2 pep/HLA-A2. The study has direct clinical relevance as the two TCRs represent both private and public responses to two different peptide antigens. This is a very thorough study as it conducts T cell assays to correlate structural features with function and specificity of the SARS-CoV-2 peptides. This includes, importantly, the testing of natural peptide variants from either SARS-CoV-2 or orthologs of the spike protein in other viruses. The public TCR YLQ7 revealed two key residues, one in the V α and one in the V β , that could explain the prevalence of the use of the TRAV12 gene and the TRBV7 gene among different individuals. One observation that does not have a clear explanation is that a region of the C α exhibited an unusual conformation in the bound vs unbound state. Aside from this, the structures and atomic details of these complexes are not unexpected but the details of the interactions are very important as they relate to T cell immunity (CD8 driven) against SARS-CoV-2. I had only a few questions/comments that could be addressed:

- It would be of value for the authors to comment on whether the higher affinity of TCR YLQ7 might be related to it being a public TCR (i.e. do other TCRs with these TRAV and TRBV have higher affinity)?
- What might explain the higher SD50 value (lower sensitivity, SD50 5.0 μ M) of T cells with the highest affinity, TLQ7 (KD 1.8 μ M), especially compared to the SD50 value of the lowest affinity TCR examined (RLQ7, KD 66 μ M, SD50 0.26 μ M)
- Related to these affinities, do the authors know if the highest affinity TCR TLQ7 is CD8 independent in Jurkat cells? This would provide additional support for it having higher functional affinity.
- The final sentence of the discussion suggests these studies can help guide the design of improved vaccines. It would be useful to provide an example.
- Page 18, I think that Fig 6d is supposed to be Fig 7d

Title: Structural basis for recognition of two HLA-A2-restricted SARS-CoV-2 spike epitopes by public and private T cell receptors

Ref.: NCOMMS-21-27572

Daichao Wu, Alexander Kolesnikov, Rui Yin, Johnathan D. Guest, Ragul Gowthaman, Anton Shmelev, Yana Serdyuk, Grigory A. Efimov, Brian G. Pierce and Roy A. Mariuzza

Reply to Reviewer #1

We thank the reviewer for his/her careful reading of our manuscript and for raising important points to address (changes in the manuscript text file are indicated with yellow highlighting):

1. The affinity between RLQ3 and RLQ/A2 is reported as mid micromolar, which is weak for alpha beta TCRs affinity for viral pMHCs, and significantly weaker than two of the other RLQ-binding TCRs. Ideally, the authors would show additional evidence that the alpha-beta pairs are correct, not inadvertently mismatched.

a. Towards this goal, showing the TRAV and TRBV frequency data from the patient sample in a supplement could be helpful.

b. Alternatively, if it is available, single cell analysis would validate the pairing.

Response: Although the affinity of TCR RLQ3 for RLQ–HLA-A2 is mid-micromolar ($K_D = 32.9 \mu\text{M}$), it is still within the affinity range ($K_D = 1\text{--}50 \mu\text{M}$) of TCRs with high functional avidity for microbial antigens (ref. 42). It should also be noted that 3D affinities, as measured by SPR, do not always correlate well with T cell activation. 2D affinities, as measured by micropipette adhesion or biomembrane force probe assays, are more reliable predictors of T cell responses (Huang et al. (2010) *Nature* **464**, 932-936). Although beyond the scope of this study, it might be illuminating to measure 2D affinities for the SARS-CoV-2-specific TCRs reported here.

The α and β chains of TCRs RLQ3, RLQ5, RLQ7, and RLQ8 were paired based on their relative frequency in MHC tetramer⁺ samples obtained from the same patients. We have included a supplementary figure illustrating the frequencies of RLQ-specific α and β chains (Fig. S1). The MHC tetramer⁺ sample from patient p1434 had three RLQ-specific α and three β chains (defined by the significant enrichment in the tetramer⁺ fraction), suggesting three RLQ-specific TCRs. We then cloned and tested four possible combinations of two top α and β chains which had to contain at least one functional pair. The only combination which was stained with MHC tetramer was RLQ3.

2. While overall reasonably conducted, there are aspects of the SPR in Figures 1 and S1 that could be improved:

a. The binding kinetics of many of the measured interactions (other than RQL5 to RQL/HLA-A2) are either too fast or approaching too fast to be measured accurately by the instrument ($\sim 0.5\text{s}^{-1}$ for koff). Therefore, the authors should show the residuals fitting of the kinetic parameters. It would also be reasonable to only report equilibrium measurements, as the kinetics do not factor into analysis of the paper.

Response: We agree that binding kinetics are too fast to be measured accurately, except for the interaction of TCR RLQ5 with RLQ–HLA-A2 (Supplementary Fig. 1a). Accordingly, we have removed information on the binding kinetics of TCRs RLQ3, RLQ8, and YLQ7 from both the text and figures.

b. Several of the equilibrium measurements (Fig 1b, 1h, S1c,d,f) do not have a sufficient number of high concentration datapoints to accurately determine an equilibrium K_d (either 0 or 1 points

over the calculated K_d , which is insufficient for a high-confidence curve fit). The authors should either re-measure the values, state them as N.D. or as greater than some value.

Response: As requested, we re-measured K_{DS} for the binding of RLQ3 to T1006I–HLA-A2 (new Fig. 1b), of YLQ7 to P272L–HLA-A2 (new Fig. 1g), of RLQ7 to RLQ–HLA-A2 (new Supplementary Fig. 1c), and of RLQ7 to T1006I–HLA-A2 (new Supplementary Fig. 1d). In each case, we used higher TCR concentrations than previously in order to reach equilibrium. We were unable to re-measure the binding of RLQ8 to T1006I–HLA-A2 (Supplementary Fig. 1f) at higher TCR concentrations because the yield of RLQ8 from in vitro folding is exceptionally low. Consequently, we report the affinity of this interaction as $K_D > 50 \mu\text{M}$.

c. Some of the sensograms for the T1006I measurements do not come to equilibrium, suggesting non-specific binding. If possible, these measurements should be redone.

Response: As noted above, we re-measured K_{DS} for the binding of RLQ3 to T1006I–HLA-A2 (new Fig. 1b) and of RLQ7 to T1006I–HLA-A2 (new Supplementary Fig. 1d).

d. Authors should include the flowrates used for the kinetic experiments and the dissociation time post injection.

Response: We have added this information to Methods.

3. I personally found the inclusion of the Rosetta-based analyses not valuable without any external validation. This is made clear by the fact that one of the Rosetta measurements with experimental validation (assessing the P272L mutation) does not match the experimental observations. However, these results are also not particularly central to the paper: I would consider de-emphasizing these because they may add more noise than signal to the work, or alternatively further buttress them with further experimental validation. This would be onerous in terms of making proteins, but would be relatively straightforward to functionally test on the reporter cell lines at least in case of the peptide mutations.

Response: As requested by the reviewer, we have de-emphasized Rosetta results by reducing the associated Results text to one paragraph per complex. The description of computational modeling results for the epitope variants is now smaller relative to the description of epitope sequence changes and experimental measurements. We have largely restricted the presentation of Rosetta results to those that directly support the crystal structures or that help us better understand the structures.

The reviewer is correct that the Rosetta prediction for TCR YLQ7 binding to the P272L peptide does not match the SPR affinity measurement. However, computational and experimental results agree reasonably well for 9 other peptides tested: MERS-RLT, HKU1-RLT, NL63-RLA, Zhejiang2013, MERS-KLQ, SARS-YLK, HKU1-PLS, OC43-PLT, and GD_pangolin. Thus, although Rosetta predictions of TCR binding are certainly not 100% reliable, we believe they do have value, especially because experimental approaches alone cannot keep pace with the vast number of existing and emerging coronavirus sequences that require evaluation. At the same time, we fully recognize that improved computational protocols are needed to reliably assess binding effects of certain substitutions.

4. I found the paper, at times, confusing to read. This was not a problem on a micro level (each part was well written), but rather in the macro sense: the early paper describes YLQ before RLQ, before switching order at the structures. The paper also describes RLQ in detail before YLQ, essentially requiring the reader to work through figures 3-6 twice. Some restructuring would help the readers out a lot. Along with this, consistent coloring for any given structure between figures,

and better labeling of the overall structures within a figure (making it more clear which structure is which in Figures 4-7) would be helpful.

Response: We agree with the reviewer's suggestions for improving the presentation. Accordingly, in the section "Interaction of SARS-CoV-2-specific TCRs with spike epitopes and epitope variants" (pp. 5-8), we now describe RLQ before YLQ, rather than the reverse, in order to match the order in the rest of the manuscript. We have also modified the figures as suggested.

5. Adding virus names to Figure 2 so the reader is not left with just the amino acid sequences would be helpful.

Response: We have added virus names to Fig. 2 as suggested.

6. Fig 3A: The difference in R_{work} vs R_{free} is 7% for RLQ/A2 structure; additionally, despite the moderate resolution of the structure the R_{work} is 0.201. Could this be evidence of overfitting?

Response: We have re-refined the coordinates of the RLQ-HLA-A2 structure. R_{free} and R_{work} are now 0.269 and 0.211, respectively. We have updated Supplementary Table 4 and PDB entry 7N1B accordingly.

7. I believe the last figure reference before the discussion (pg 18) should reference 7d rather than 6d.

Response: Thank you for catching this error.

8. The structural basis of public responses is a quite interesting topic – the authors discuss this but do not have it clearly laid out in one figure. Collating a figure that directly shows the basis for these V regions would be interesting (either for main text or supplement).

Response: As suggested, we have added Supplementary Fig. 6 to illustrate the structural basis for the TRAV and TRBV germline gene bias of TCR YLQ7.

Title: Structural basis for recognition of two HLA-A2-restricted SARS-CoV-2 spike epitopes by public and private T cell receptors

Ref.: NCOMMS-21-27572

Daichao Wu, Alexander Kolesnikov, Rui Yin, Johnathan D. Guest, Ragul Gowthaman, Anton Shmelev, Yana Serdyuk, Grigory A. Efimov, Brian G. Pierce and Roy A. Mariuzza

Reply to Reviewer #2

We are grateful to the reviewer for his/her appreciation of our study and for calling our attention to the following points (changes in the manuscript text file are indicated with yellow highlighting):

- It would be of value for the authors to comment on whether the higher affinity of TCR YLQ7 might be related to it being a public TCR (i.e. do other TCRs with these TRAV and TRBV have higher affinity)?

Response: As far as we are aware, public TCRs do not have generally higher affinities than private TCRs. While it is true that public TCR YLQ7 ($K_D = 1.8 \mu\text{M}$) has higher affinity than private TCR RLQ3 ($32.9 \mu\text{M}$), another private TCR, RLQ5, has an affinity ($3.4 \mu\text{M}$) that is essentially indistinguishable from that of YLQ7.

- What might explain the higher SD50 value (lower sensitivity, SD50 $5.0 \mu\text{M}$) of T cells with the highest affinity, TLQ7 ($K_D 1.8 \mu\text{M}$), especially compared to the SD50 value of the lowest affinity TCR examined (RLQ7, $K_D 66 \mu\text{M}$, SD50 $0.26 \mu\text{M}$).

Response: We do not have a simple explanation for these results, except to point out that 3D affinities, as measured by SPR, do not always correlate well with T cell activation (EC_{50}). 2D affinities, as measured by micropipette adhesion or biomembrane force probe assays, are more reliable predictors of T cell responses (Huang et al. (2010) *Nature* **464**, 932-936). Although beyond the scope of this study, it might be illuminating to measure 2D affinities for the SARS-CoV-2-specific TCRs reported here.

- Related to these affinities, do the authors know if the highest affinity TCR TLQ7 is CD8 independent in Jurkat cells? This would provide additional support for it having higher functional affinity.

Response: We thank the reviewer for the suggestion to further investigate the functional avidity of TCR YLQ7 and its dependence on binding by CD8 co-receptor. In the process, we made two new Jurkat reporter lines (with and without CD8 co-receptor) and repeated the peptide titration experiments multiple times. The new YLQ7-CD8 line was made by sorting CD8⁺ cells (in the same way as we made the RLQ3, RLQ5, RLQ7, and RLQ8 lines) instead of MHC tetramer sorting, as was done for the YLQ7 line previously. We discovered that the new line not only displayed a functional avidity of $0.4 \mu\text{M}$, which was in more accordance with the SPR data, but also was sensitive to the P272L substitution. This is in agreement with recent data from Andrew Sewell's group on the functional significance of the P272L substitution (new ref. 47). We believe that these new results are more physiological and have replaced the related figures and results in the manuscript.

Regarding the CD8 contribution to TCR-pMHC binding, we performed peptide titration experiments with CD8 blocking antibodies and with the YLQ7 reporter line without CD8 co-receptor (new Supplementary Fig. 3). Both results indicate only a modest (6-fold) reduction of the

functional avidity which is consistent with previously reported data on the high affinity virus-specific TCRs.

- The final sentence of the discussion suggests these studies can help guide the design of improved vaccines. It would be useful to provide an example.

Response: As an example of structure-guided T cell vaccine design, we cite a recent article describing the application of structure-based network analysis to identify mutationally constrained CD8⁺ T cell epitopes that are conserved across SARS-CoV-2 variants and sarbecoviruses (Nathan et al. (2021) *Cell* **184**, 4401-4413; new ref. 69).

- Page 18, I think that Fig 6d is supposed to be Fig 7d.

Response: We have made this correction.

REVIEWER COMMENTS

Reviewer #1 (Remarks to the Author):

I thank the authors for the work on this resubmission - I think the paper is substantially improved and clearer to read. My remaining concern is the issue of TCR chain pairing (as brought up in point 1 of my initial review). The relative closeness of the observed alpha and beta frequencies shown in figure S1, and the fact that RLQ1, 2, and 4 show no activity on the reporter line assay, provide context for this worry.

That being said, I understand that the authors do not have samples to return to in order to ensure these chain pairings (as they were obtained from previously published sequenced samples). Papers describing sequences preceding structural analyses is relatively normal for TCR structural examinations, but so are functional yet non-natural chain pairings (the AND TCR, a mouse class II restricted TCR which uses the AN6.2 alpha chain and the 5C.C7 beta chain, is maybe the best studied example).

Despite this inherent limitation of the study, I support publication as this is a collection of well-conducted and timely work. I suggest three revisions to the text (and no further experiments) before publication:

1 - the authors should make the fact that the RLQ TCRs derived from previous sequencing data more explicit. Right now, the start of the results section is ambiguous about whether the technique was conducted as described in ref. 26, or the literal data was used. It's the latter, which is totally fine, but it should be easier to track that.

2 - I would include information of the non-functional TCRs in Fig. S1 in Table S1, to better contextualize the study.

3- I would explicitly mention the method of pairing the RLQ TCRs as a limitation of the study (probably in the discussion) for better context for any non-expert readers of the paper.

Minor point - the electron density for RLQ3 in Figure S5 did not show up for me, probably some rendering error.

Reviewer #2 (Remarks to the Author):

The revision has addressed all of my original comments

Title: Structural assessment of HLA-A2-restricted SARS-CoV-2 spike epitopes recognised by public and private T-cell receptors

Ref.: NCOMMS-21-27572A

Daichao Wu, Alexander Kolesnikov, Rui Yin, Johnathan D. Guest, Ragul Gowthaman, Anton Shmelev, Yana Serdyuk, Grigory A. Efimov, Brian G. Pierce and Roy A. Mariuzza

Reply to Reviewer #1

We appreciate the reviewer's concerns about TCR α/β chain pairing and have addressed them as follows (changes in the manuscript text file are indicated with yellow highlighting):

1 - The authors should make the fact that the RLQ TCRs derived from previous sequencing data more explicit. Right now, the start of the results section is ambiguous about whether the technique was conducted as described in ref. 26, or the literal data was used. It's the latter, which is totally fine, but it should be easier to track that.

Response: We agree that the source of the TCR sequences is ambiguous. We have modified the text for clarity (p. 5, last paragraph):

“Sequences of α and β chains for YLQ- and RLQ-specific TCRs were obtained from a previous study (26).”

2 - I would include information of the non-functional TCRs in Fig. S1 in Table S1, to better contextualize the study.

Response: As requested, we have added information on non-functional TCRs to Supplementary Table 1.

3- I would explicitly mention the method of pairing the RLQ TCRs as a limitation of the study (probably in the discussion) for better context for any non-expert readers of the paper.

Response: We agree and have added the following paragraph to the Discussion (p. 18, 2nd paragraph):

“A possible limitation of this study is the frequency-based pairing of TCR α and β chains. We therefore cannot formally exclude that some RLQ-specific TCRs might be unnaturally paired, even though they bound RLQ–HLA-A2 with affinities characteristic of TCRs with high functional avidity for microbial antigens ($K_D < 50 \mu\text{M}$) (42). By contrast, the natural pairing of YLQ7 α and β chains was independently confirmed in another study using single-cell sequencing (41).”

Minor point - the electron density for RLQ3 in Figure S5 did not show up for me, probably some rendering error.

Response: I have rechecked Figure S5 as submitted to the journal. It shows electron density for RLQ3, at least on my desktop.